# Structural insights into the π-π-π stacking mechanism and DNA-binding activity of the YEATS domain

Brianna J. Klein[1], Kendra R. Vann[1], Forest H. Andrews[1], Wesley W. Wang[2], Jibo Zhang[3], Yi Zhang [1], Anastasia A. Beloglazkina[4], Wenyi Mi [5], Yuanyuan Li[6], Haitao Li [6], Xiaobing Shi [5], Andrei G. Kutateladze[4], Brian D. Strahl [3], Wenshe R. Liu [2] & Tatiana G. Kutateladze [1]

The YEATS domain has been identified as a reader of histone acylation and more recently emerged as a promising anti-cancer therapeutic target. Here, we detail the structural mechanisms for π-π-π stacking involving the YEATS domains of yeast Taf14 and human AF9 and acylated histone H3 peptides and explore DNA-binding activities of these domains. Taf14-YEATS selects for crotonyllysine, forming π stacking with both the crotonyl amide and the alkene moiety, whereas AF9-YEATS exhibits comparable affinities to saturated and unsaturated acyllysines, engaging them through π stacking with the acyl amide. Importantly, AF9-YEATS is capable of binding to DNA, whereas Taf14-YEATS is not. Using a structure-guided approach, we engineered a mutant of Taf14-YEATS that engages crotonyllysine through the aromatic-aliphatic-aromatic π stacking and shows high selectivity for the crotonyl H3K9 modification. Our findings shed light on the molecular principles underlying recognition of acyllysine marks and reveal a previously unidentified DNA-binding activity of AF9-YEATS.

---

[1] Department of Pharmacology, University of Colorado School of Medicine, Aurora, CO 80045, USA. [2] Department of Chemistry, Texas A&M University, College Station, TX 77843, USA. [3] Department of Biochemistry & Biophysics, The University of North Carolina School of Medicine, Chapel Hill, NC 27599, USA. [4] Department of Chemistry and Biochemistry, University of Denver, Denver, CO 80210, USA. [5] Center for Epigenetics, Van Andel Research Institute, Grand Rapids, MI 49503, USA. [6] Department of Basic Medical Sciences, School of Medicine, Tsinghua University, 100084 Beijing, P.R. China. These authors contributed equally: Brianna J. Klein, Kendra R. Vann, Forest H. Andrews. Correspondence and requests for materials should be addressed to T.G.K. (email: tatiana.kutateladze@ucdenver.edu)

A large number of epigenetic marks or posttranslational modifications (PTMs) in histones have been discovered over the past few years[1,2]. One of the major and widespread PTMs is acylation of the ε-amino group of lysine residues. Acylation neutralizes the positive charge and increases the hydrophobic character of the lysine side chain and alters chromatin structure utilizing two fundamental mechanisms. It weakens nonspecific electrostatic interactions between histones and DNA, leading to a more open and transcriptionally active chromatin. It also recruits acyllysine readers and their host proteins and complexes to specific genomic loci to facilitate diverse epigenetic-driven nuclear programs essential in chromatin remodeling, gene transcription, and DNA replication, recombination and repair[3,4]. At least nine acyllysine modifications in histones have been identified, including crotonylation.

Since the discovery of lysine crotonylation in mammalian cells, this PTM has drawn much attention becoming an increasingly important epigenetic mark[5,6]. Crotonylation is enriched around active gene promoters and potentially enhancers and was found to stimulate gene transcription to a higher degree than the corresponding acetylation[6]. Studies of genomic distribution of crotonyllysine and acetyllysine reveal some differences, which suggest that these modifications are not redundant and can be associated with distinct biological outcomes[5,6]. Nevertheless, a substantial overlap in genomic localization and similarities in chemical properties present a challenge in studying and distinguishing the biological roles of these PTMs. A strategy of employing separate readers as probes for targeting individual acyllysine modifications has not been well developed, in part because of the promiscuous nature of currently known readers. Canonical acyllysine readers, bromodomain and double PHD finger (DPF), display comparable binding capabilities toward various short acyl chain modifications[7–12], and while the YEATS domain exhibits preference for crotonyllysine, it still associates albeit more weakly with other acyllysine marks[13–22]. We have previously shown that the YEATS domains of Taf14 (Taf14-YEATS) and AF9 (AF9-YEATS) recognize crotonyllysine through a non-canonical π-π-π stacking mechanism[16,17], and in the case of Taf14-YEATS it involves aromatic-amide-aromatic π stacking and aromatic-aliphatic-aromatic π stacking[16] (Fig. 1a). In this study, we elucidate the significance of π stacking components in targeting of acetyllysines by the YEATS domains and report a previously uncharacterized DNA-binding activity of the human AF9-YEATS domain. We also show how the unique nature of the conjugated π system of the crotonyl modification could aid in the design of effectors indifferent to saturated acyllysines.

## Results and discussion

**Structural insight into the selectivity of Taf14-YEATS.** To determine the contribution of π stacking to the binding energetics, we measured binding affinities of Taf14-YEATS to the histone H3K9acyl peptides containing unsaturated and saturated four-carbon acyl modifications. Tryptophan fluorescence measurements showed that binding of Taf14-YEATS was reduced ~3-fold when the crotonylated modification was replaced with a similar in length but saturated butyryl modification (Figs. 1b, c and Supplementary Figure 1). The binding was further decreased to a negatively charged succinyllysine, and no interaction was detected with a branched hydroxyisobutyryllysine in $^1$H,$^{15}$N heteronuclear single quantum coherence (HSQC) NMR titration experiments (Supplementary Figure 2). The selectivity of the YEATS domain for the crotonyllysine modification was substantiated by pulldown assays with biotinylated histone peptides (Fig. 1d). GST-fused Taf14-YEATS recognized the H3K9cr peptide but associated weaker with H3K9bu and H3K9ac peptides.

Together these results show that Taf14-YEATS selects for unbranched acyl modifications, and the alkene group in the modification enhances the protein-binding capability.

To gain insight into the molecular basis for the enhancement, we obtained the crystal structure of Taf14-YEATS in complex with H3K9bu peptide and compared it with the previously reported structure of this domain in complex with H3K9cr[16]. The structures were superimposed with a root mean square deviation of 0.1 Å, indicating overall conservation of the binding mode. As in the case of crotonyllysine, butyryllysine transversed a narrow tunnel and was sandwiched between two aromatic residues of the protein, W81 and F62 (Fig. 1e, Supplementary Figure 3a and Supplementary Table 1). Amide nitrogen and carbonyl oxygen of butyryllysine were restrained through hydrogen bonds with the hydroxyl group of T61 and backbone amide of W81 and through a water-mediated hydrogen bond with G82. A similar set of polar interactions stabilized the complex with crotonyllysine, however, substantial differences were observed in the position of W81. It adopted a single conformation, rotamer 1 (r1), involved in the π stacking interaction with butyryl amide in the Taf14 YEATS–H3K9bu complex. In contrast, when complexed with H3K9cr, W81 adopted two conformations, with r1 and r2 providing maximum π stacking with the crotonyl amide and the crotonyl alkene moiety, respectively (Supplementary Figures 3b and 4). Neither the complex with propionylated H3K9pr peptide nor the complex with acetylated H3K9ac peptide had the r2 conformation of W81, indicating that it is a unique feature available to the unsaturated acyl modification but not to saturated acyllysine modifications (Fig. 1f).

**Design of the crotonyllysine-specific Taf14-YEATS.** Comparison of the H3K9bu- and H3K9cr-bound Taf14-YEATS structures revealed that the side chain of W81 is located within 4 Å of the α carbon atom of G82. Modeling in G82A mutation suggested that the presence of an additional methyl group at position 82 would hinder the ability of W81 to adopt the r1 conformation required for the π amide stacking. To test this hypothesis, we generated the G82A mutant of Taf14-YEATS and determined its crystal structure (Fig. 2a). In support of our idea, the indole moiety of W81 in this mutant was in a tilted conformation incompatible with r1 (Fig. 2b). We next assessed the acyllysine-binding activity of Taf14-YEATS G82A using NMR titration experiments (Fig. 2c). Large chemical shift perturbations (CSPs) in $^{15}$N-labeled Taf14-YEATS G82A upon addition of H3K9cr peptide indicated that the mutant retains its ability to recognize the crotonyl modification. However, association of Taf14-YEATS with the H3K9ac peptide was almost negligible, as very small CSPs were observed (Fig. 2c and Supplementary Figure 5b). Tryptophan fluorescence measurements yielded a $K_d$ of 124 μM for the interaction of Taf14-YEATS G82A with H3K9cr (Fig. 2d). Although binding affinity of the G82A mutant to H3K9cr was reduced compared with the binding affinity of the wild-type protein, it remains in the range of affinities exhibited by the well-established readers of acetyllysine, bromodomains[7,9].

We next characterized the Taf14-YEATS G82A mutant in the cellular context. We have previously shown that abrogating the Taf14 association with H3K9acyl through mutating W81 to alanine impacts the transcript levels of a variety of yeast genes both positively and negatively[14]. To determine the effect of blocking Taf14 from selectively associating with H3K9ac but not with H3K9cr, we carried out a real-time quantitative PCR (qPCR) analysis for a set of Taf14-regulated genes in the *TAF14* deletion strain (*taf14Δ*) rescued with a vector only, wild-type *TAF14*, or with a form carrying either the W81A or G82A mutation in the YEATS domain (*taf14* W81A and *taf14* G82A) (Fig. 2e). As

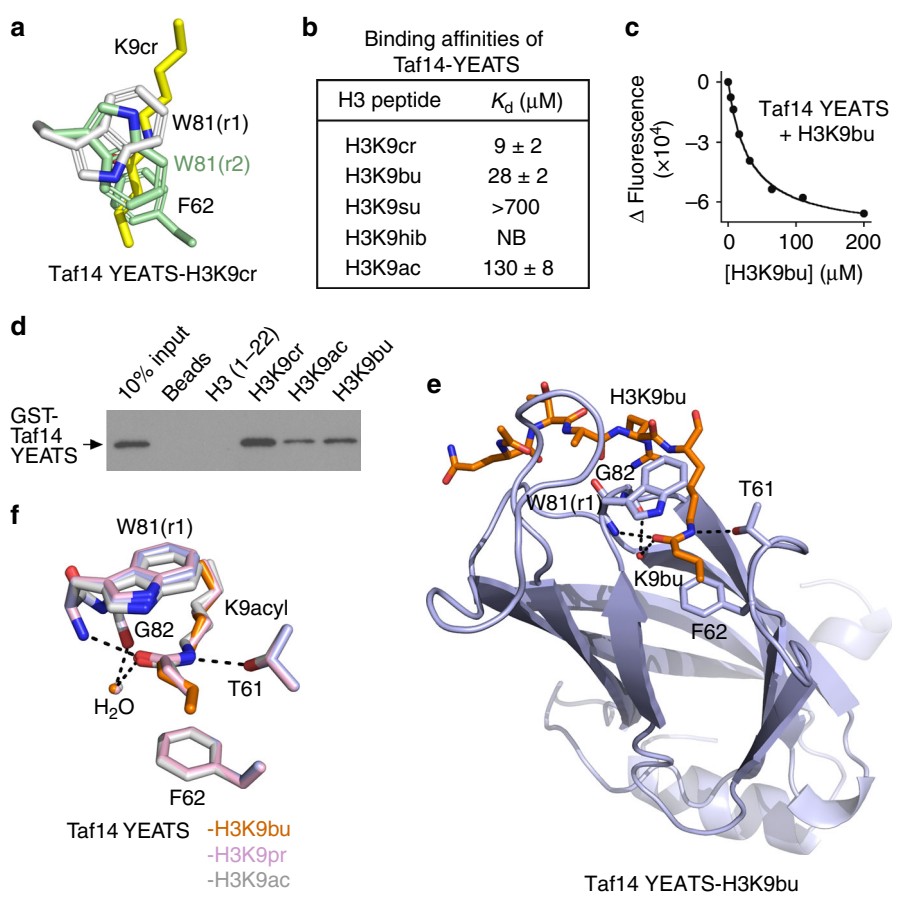

**Fig. 1** Structural insight into the selectivity of Taf14-YEATS. **a** Crotonylated lysine (yellow) is sandwiched between W81 and F62 in the complex of Taf14-YEATS with H3K9cr. W81 adopts two conformations, rotamer 1 (light gray) and rotamer 2 (green). **b** Binding affinities of Taf14-YEATS to the indicated histone peptides, as measured by fluorescence (cr, bu, ac) or NMR (su, hib). Values represent the average of three separate experiments (two for H3K9ac) with error calculated as the SD between the runs. The enhancement in selectivity of Taf14-YEATS to crotonyllysine is comparable to the enhancement in selectivity of other well-recognized epigenetic readers, such as DPFs[8,11]. **c** Representative binding curves used to determine $K_d$ by tryptophan fluorescence. **d** Peptide pulldown assays for Taf14-YEATS using indicated histone H3 peptides. **e** The ribbon diagram of the Taf14-YEATS:H3K9bu complex. Dashed lines and red sphere represent hydrogen bonds and a water molecule, respectively. The YEATS domain is colored lavender and H3K9bu peptide is colored orange. Residues of the YEATS domain involved in the binding of K9bu are labeled. **f** Structural overlay of the acyllysine binding sites in the Taf14-YEATS: H3K9bu (lavender/orange), Taf14-YEATS:H3K9pr (pink), and Taf14-YEATS:H3K9ac[14] (gray) complexes

expected, *taf14Δ* deletion impacted the expression of multiple genes that were previously reported to be regulated by Taf14 interaction with H3K9acyl, including *YBL041, YPR145, YPR158, YER145C, YKL150W*, and *YKR093W*[14]. For control, we also included several genes, such as *YLR290, YAL017*, and *MRPL10*, which were shown not to be regulated by Taf14[14]. Intriguingly, although the effect of the entirely loss-of-function *taf14* W81A mutant was largely consistent with the effect seen with the loss of Taf14, the *taf14* G82A mutant showed more variable results (Fig. 2e). In some cases, the *taf14* G82A mutant impacted the expression of genes similar to that of the *taf14* W81A mutant (*YBL041, YPR158, YKL150W*, and *YKR093W*), whereas in other cases, the *taf14* G82A mutant showed distinct effects on gene transcripts not seen in either the *taf14* W81A mutant or the *TAF14* deletion (*YPR145* and *YER145C*). Given the *taf14* G82A mutation impacts H3K9ac binding to a greater degree than H3K9cr binding, these findings suggest a differential requirement of H3K9ac and H3K9cr for the proper expression of Taf14-regulated genes.

**Crotonyllysine recognition by Taf14-YEATS G82A.** To elucidate the mechanism by which Taf14-YEATS G82A distinguishes

crotonyllysine and discriminates against saturated counterparts, we obtained the crystal structure of the H3K9cr-bound G82A mutant (Fig. 3a). The structure of the complex showed that the aromatic side chains of W81 and F62 lay parallel to each other and at equal distance of 3.4–3.6 Å from the crotonyl alkene group (Fig. 3b). Remarkably, only a single r2 conformation of W81 was observed in the structure of the complex. Analysis of molecular orbitals (MOs) for the W81, Kcr, and F62 assembly at the B3LYP/6-311 + G(d,p) level of density functional theory (DFT) revealed that the bonding MOs transcending all three π systems are deeply buried, indicating considerable synergism of the π-π-π stacking interaction (Supplementary Figure 6). A representative MO, the highest occupied MO HOMO-8, is shown in Fig. 3c. We concluded that the ability of the W81 r2 rotamer to engage the crotonyl alkene moiety most likely accounts for the selectivity of Taf14-YEATS G82A.

**AF9-YEATS is a more versatile acyllysine reader.** Alignment of the YEATS domain sequences derived from the family of human and yeast proteins shows conservation of the WG motif (W81 and G82 in Taf14) in all YEATS members but AF9 and ENL, which instead contain a YA motif (Fig. 4a). Interestingly, despite

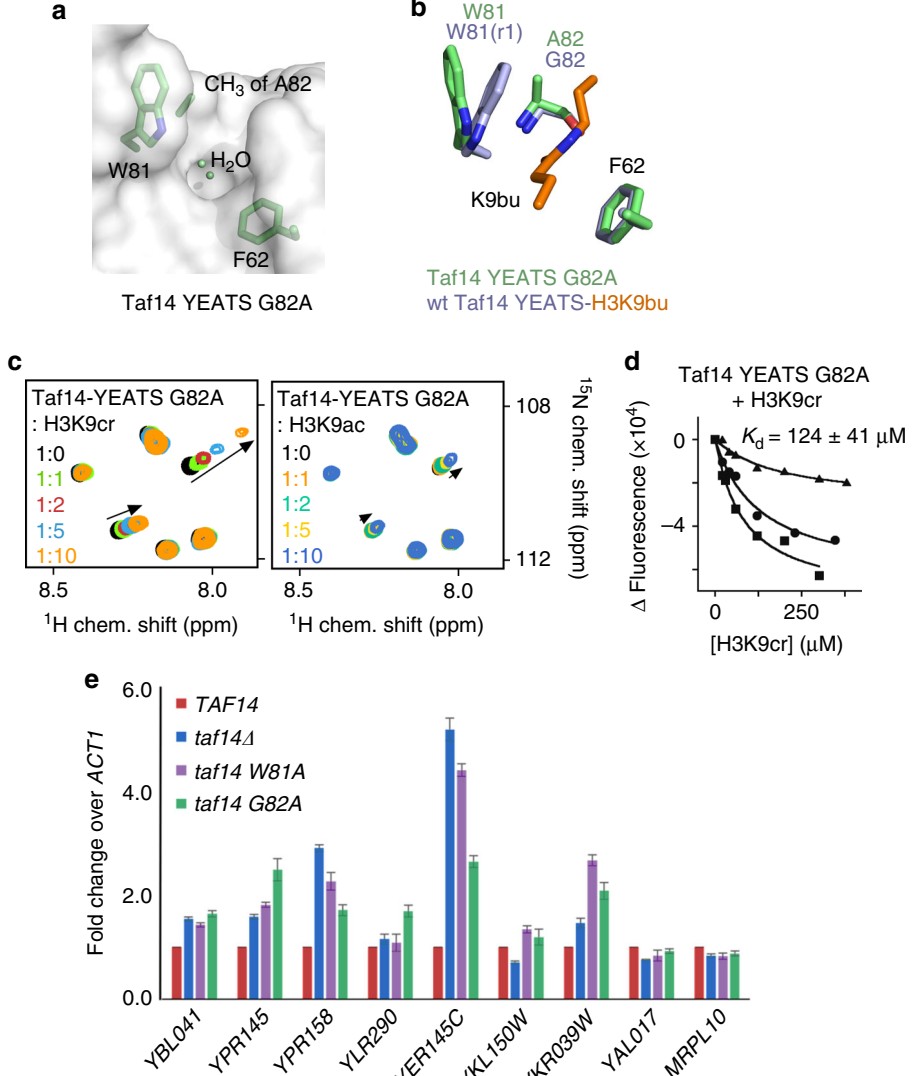

**Fig. 2** Engineering the Taf14-YEATS-based reader of H3K9cr. **a** Surface view of the Taf14-YEATS G82A mutant structure in the apo-state. The side chains of W81, A82, and F62 are shown as green sticks. **b** Structural overlay of the acyllysine-binding site in the Taf14-YEATS:H3K9bu (lavender/orange) complex and the apo state of Taf14-YEATS G82A (green). **c** Superimposed $^1$H,$^{15}$N HSQC spectra of uniformly $^{15}$N-labeled G82A mutant of Taf14-YEATS recorded while the indicated peptides were added stepwise. The spectra are color coded according to the protein:peptide molar ratio. **d** Binding curves used to determine $K_d$ of taf14-YEATS G82A by tryptophan fluorescence. **e** Real-time qPCR analysis of various transcripts in the wild-type strain, *TAF14* delete strain, and *taf14* mutant strains. The mean ± SD are calculated from three biological replicates

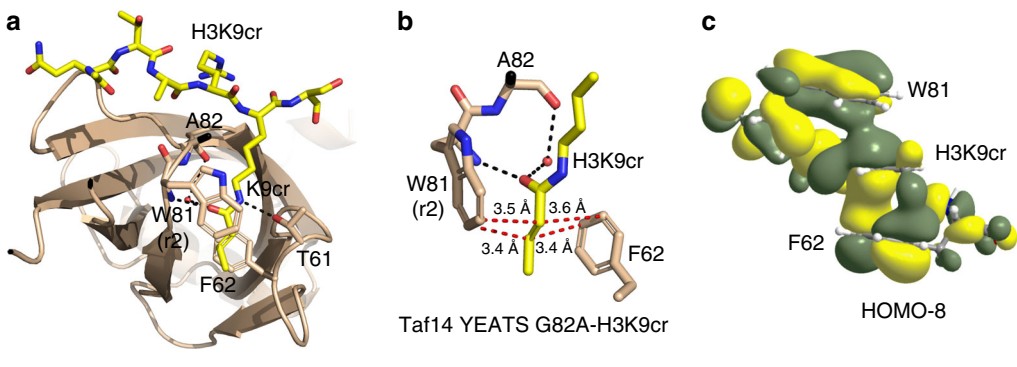

**Fig. 3** Structural basis for recognition of crotonyllysine by Taf14-YEATS G82A. **a** The ribbon diagram of the Taf14-YEATS G82A:H3K9cr complex. Dashed lines and red sphere represent hydrogen bonds and a water molecule, respectively. **b** Close up view of the H3K9cr-binding site of Taf14-YEATS G82A. Red dashed lines represent short distances indicative of the aromatic-aliphatic-aromatic π stacking interaction. **c** A representative HOMO-8 (see also Supplementary Figure 6)

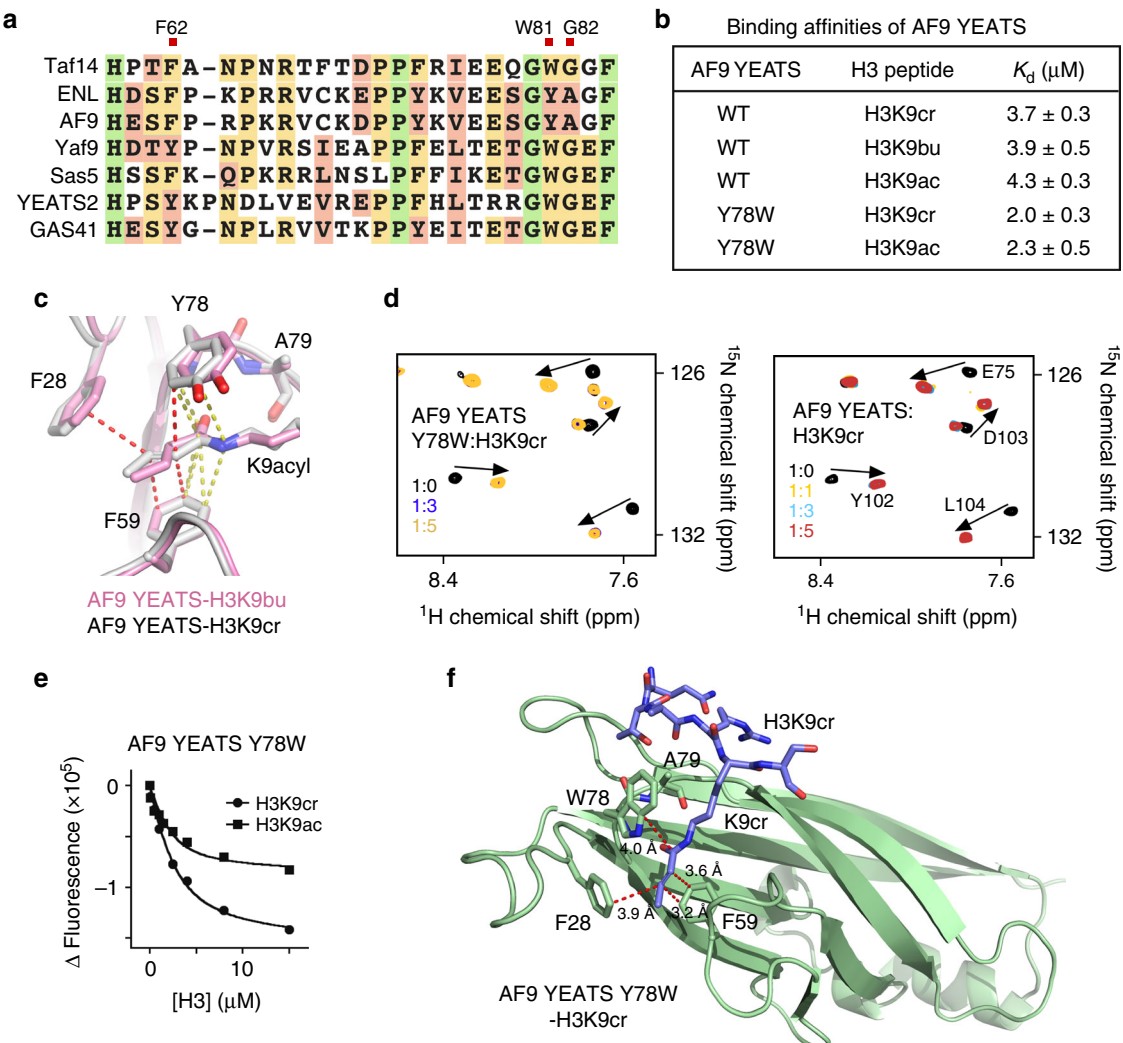

**Fig. 4** Structural insight into the selectivity of AF9-YEATS. **a** Alignment of the YEATS domain sequences: absolutely, moderately, and weakly conserved residues are colored green, orange, and pink, respectively. **b** Binding affinities of AF9-YEATS to indicated histone peptides. Values represent the average of three separate experiments with error calculated as the SD between the runs. **c** Structural overlay of the acyllysine-binding sites in the AF9-YEATS:H3K9bu (pink) and AF9-YEATS:H3K9cr[17] (light gray) complexes. Yellow dashed lines represent short distances indicative of the aromatic-amide-aromatic π stacking interaction in AF9-YEATS:H3K9cr[17]. Red dashed lines represent short, <4 Å, distances between the crotonyl alkene group and the aromatic residues. **d** Superimposed ¹H,¹⁵N HSQC spectra of AF9-YEATS, wt and Y78W mutant, recorded in the presence of increasing concentration of H3K9cr peptide. Spectra are color coded according to the protein:peptide molar ratio. **e** Representative binding curves used to determine $K_d$ of AF9-YEATS Y78W by tryptophan fluorescence. **f** The ribbon diagram of the AF9-YEATS Y78W:H3K9cr complex. Red dashed lines represent short, <4 Å, distances between the crotonyl alkene group of H3K9cr and the aromatic residues of the protein

the presence of an alanine following the aromatic Y78 residue in AF9, the YEATS domain of AF9 did not differentiate between acylated modifications on H3K9 and bound almost equally well to H3K9cr, H3K9bu, and H3K9ac, supporting the original findings[17] (Fig. 4b and Supplementary Figure 7). To compare the binding modes of the YEATS domains harboring the WG and YA motifs, we determined the structure of AF9-YEATS in complex with H3K9bu peptide. Structural overlay of the AF9-YEATS: H3K9bu and previously reported AF9-YEATS:H3K9cr[17] complexes showed that both saturated and unsaturated four-carbon acyl chains of K9 are bound by the three aromatic residues Y78, F59, and F28 in a very similar way (Fig. 4c). Short (<4 Å) distances observed between $C^\gamma$, $C^\delta$, $C^\epsilon$, and $C^\eta$ atoms of Y78, the amide group of acyllysine, and the $C^\epsilon$ and $C^\zeta$ atoms of F59 indicated that Y78 and F59 of AF9 are involved in the aromatic-amide-aromatic π-π-π stacking in either complex (Fig. 4c, yellow dashed lines). However, neither Y78 nor F28 appear to form π-π-

π stacking with the alkene group of crotonyllysine as only a single carbon atom in Y78 ($C^\epsilon$) or F28 ($C^\epsilon$) was close enough to the C = C carbon atoms to fully engage the double bond (Fig. 4c, red dashed lines). This inability to form the additional energetically favorable contact is likely the reason behind the indifference of the YEATS domain of AF9 toward crotonyl-, butyryl-, and acetyl-H3K9acyl modifications.

We next explored the possibility of engineering an AF9-based acyllysine-specific reader through mutating Y78 in AF9-YEATS to a tryptophan and therefore re-creating the WA motif seen in the Taf14-YEATS G82A mutant. Although NMR titration experiments revealed a tight interaction of AF9-YEATS Y78W with H3K9cr, binding affinities measured by fluorescence spectroscopy indicated that this mutant does not differentiate between H3K9cr and H3K9ac and associates with both peptides with a $K_d$ of 2 μM (Figs. 4b, d, e and Supplementary Figure 8a). To explain the different behavior of Taf14-YEATS G82A and

AF9-YEATS Y78W, we obtained the crystal structure of AF9-YEATS Y78W in complex with H3K9cr peptide (Fig. 4f). In this complex, W78 was in a single r1 conformation, which was engaged in π stacking with the crotonyl amide. Notably, A79 imposed steric hindrance on W78 to a higher degree than on wild-type Y78 at this position, resulting in a tilted conformation of W78 (Supplementary Figure 8b). W78 is incapable of adopting the r2 conformation necessary for π stacking with the alkene group most likely because of steric hindrance from F28 (Supplementary Figure 8c). Additional mutation of A79 to a glycine in AF9, to recapitulate the Taf14' WG motif, failed to induce the selectivity as the AF9-YEATS Y78WA79G mutant bound equally well to either H3K9cr or H3K9ac peptide (Fig. 5a). By contrast, the W81Y mutant of Taf14-YEATS retained its capability to select for H3K9cr, suggesting that position of the tyrosine residue in this mutant allowed for π stacking with the alkene group (Fig. 5a and Supplementary Figure 9).

**AF9-YEATS binds to DNA**. Analysis of electrostatic surface potential of AF9-YEATS reveals that the part of the domain opposite to the H3K9cr-binding site is highly positively charged. Particularly, two apparent clusters of basic residues on the protein surface suggest that they could be involved in binding to negatively charged DNA (Fig. 5b). To determine whether AF9-YEATS is capable of interacting with DNA, we examined association of AF9-YEATS with 147-bp 601 Widom DNA using an electrophoretic mobility shift assay (EMSA). The 601 DNA was incubated with increasing amounts of AF9-YEATS and the reaction mixtures were resolved on native polyacrylamide gels (Fig. 5c). A gradual increase in the amount of added AF9-YEATS resulted in a shift and disappearance of the 601 DNA band and the appearance of the bands corresponding to multiple complexes formed between AF9-YEATS and multiple major/minor grooves[23] of 601 DNA (Fig. 5c). Direct binding to DNA was corroborated by NMR titration experiments, in which substantial CSPs were observed in [15]N-labeled AF9-YEATS upon gradual addition of 601 DNA (Fig. 5d and Supplementary Figure 10a, b).

To assess how AF9-YEATS interacts with both H3K9acyl and DNA, we reconstituted H3K9acyl-containing nucleosome core particles (H3K9cr-NCPs and H3K9ac-NCP) and examined binding of AF9-YEATS to these NCPs and to unmodified NCP in EMSA. Incubation of increasing amounts of AF9-YEATS with H3K9cr-NCP led to a disappearance of the H3K9cr-NCP band and of free 601 DNA present in the sample, indicating formation of the AF9-YEATS:H3K9cr-NCP and AF9-YEATS:601 DNA, respectively, complexes (Fig. 5e). We note that the free 601 DNA band disappeared faster than the H3K9cr-NCP band, implying that AF9-YEATS prefers a more accessible, free DNA to the DNA wrapped around the nucleosome, despite the fact that H3K9cr-NCP contains the additional binding partner of the YEATS domain—H3K9cr. Yet, the recognition of H3K9cr by the YEATS domain is required for strong interaction, as the AF9-YEATS association with unmodified NCP was substantially compromised (Fig. 5f) and its association with H3K9ac-NCP was slightly diminished (Supplementary Figure 10c). In further support, the H3K9cr-NCP binding capability of the AF9-YEATS F59A/Y78A mutant, which is defective in H3K9acyl binding[22], was also markedly decreased (Fig. 5g).

**DNA- and H3K9cr-binding sites in AF9-YEATS do not overlap**. To map the DNA-binding site of AF9-YEATS, we plotted NMR CSPs induced in each amide of [15]N-labeled AF9-YEATS by 601 DNA and compared with CSPs induced in this protein by the H3K9cr peptide (Figs. 6a, b). Notably, residues located in and around the H3K9cr-binding site, including Y78, were most

perturbed upon addition of the H3K9cr peptide (Figs. 6a, c). However, an entirely different set of residues was perturbed due to binding of 601 DNA (Figs. 6b, c). The most pronounced changes were observed in the residues located in three AF9-YEATS regions, i.e., the 60th, 90th, and 130th patches (Figs. 6c, d). These patches contain the surface exposed positively charged R61, K63, R64, K67, K92, R96, K97, R133, K134, and K137 residues that can be directly or indirectly involved in electrostatic interactions with negatively charged 601 DNA.

To determine the contribution of the perturbed patches to DNA binding, we generated four mutants of AF9-YEATS, including R61E/K63E/K67E, K92E/R96E/K97E, R96E/K97E, and R133E/K134E/K137E. Among them, only the R61E/K63E/K67E (thereafter referred to as RKK) mutant was soluble and folded. Binding of the AF9-YEATS RKK mutant to 601 DNA and H3K9cr-NCP was examined by EMSA. As shown in Figs. 6e-g, the DNA-binding and H3K9cr-NCP-binding abilities of this mutant were markedly reduced compared with the respective binding abilities of WT AF9-YEATS, pointing to a critical role of the 60th patch residues of AF9-YEATS in the interaction with DNA (Fig. 6c). Taken together, the EMSA results and NMR CSPs indicate that the 60th patch residues are part of the DNA-binding site of AF9-YEATS. A lack of mutually perturbed residues due to the interaction with H3K9cr peptide or 601 DNA further suggests that the H3K9cr- and DNA-binding sites of AF9-YEATS do not overlap.

**Bivalent association of AF9 is not conserved in Taf14-YEATS**. To identify the DNA fragments smaller than 147 bp of 601 DNA that can be bound by AF9-YEATS, we carried out EMSA assays with 20-bp DNA and 15-bp DNA (Fig. 7a and Supplementary Figure 11a). Both 15- and 20-mer DNA bands disappeared in an AF9-YEATS concentration-dependent manner, and quantitative analysis of the band intensities yielded $K_d$s of 47 μM and 57 μM for 15-mer DNA and 20-mer DNA, respectively (Fig. 7b), which is in the range of DNA-binding affinities of other transcriptional regulators[24]. Overall, these and the results described above suggest that AF9-YEATS interacts with chromatin in a bivalent manner, i.e., it binds both H3K9acyl and DNA (Fig. 7c).

Is the bivalent interaction conserved in other YEATS domains? Alignment of the YEATS sequences reveals that all three positively charged patches are conserved in the YEATS domain of another human protein, ENL, and thus ENL-YEATS is likely capable of binding to DNA (Fig. 7d). Although these patches are not strictly conserved in the YEATS domain of the human protein YEATS2, the "DNA-binding" surface in YEATS2-YEATS is also highly positively charged (Supplementary Figure 11b). Altogether, these observations suggest that the DNA-binding function may be conserved in the family of human YEATS readers. In contrast, the corresponding Taf14-YEATS surface is enriched in the negatively charged residues that can electrostatically repulse DNA (Fig. 7e). Indeed, EMSA experiment using 601 DNA showed that even a large, up to ~10[3]-fold, excess of Taf14-YEATS failed to shift the DNA band, implying that this domain does not bind DNA (Fig. 7f). Interestingly, similar to Taf14-YEATS, the YEATS domain of another yeast protein, Yaf9, also has a negatively charged and therefore repulsive of DNA surface opposite to its H3K27acyl-binding site, which may indicate that the ability to interact with DNA is lost in yeast YEATS domains (Supplementary Figure 11b).

Collectively, our results have illuminated the mechanistic basis underlying distinct recognition of acyllysine PTMs by the YEATS domains of yeast Taf14 and human AF9, allowing us to develop a Taf14-YEATS module that selects for the crotonyl modification and discriminates against saturated acyl modifications. The

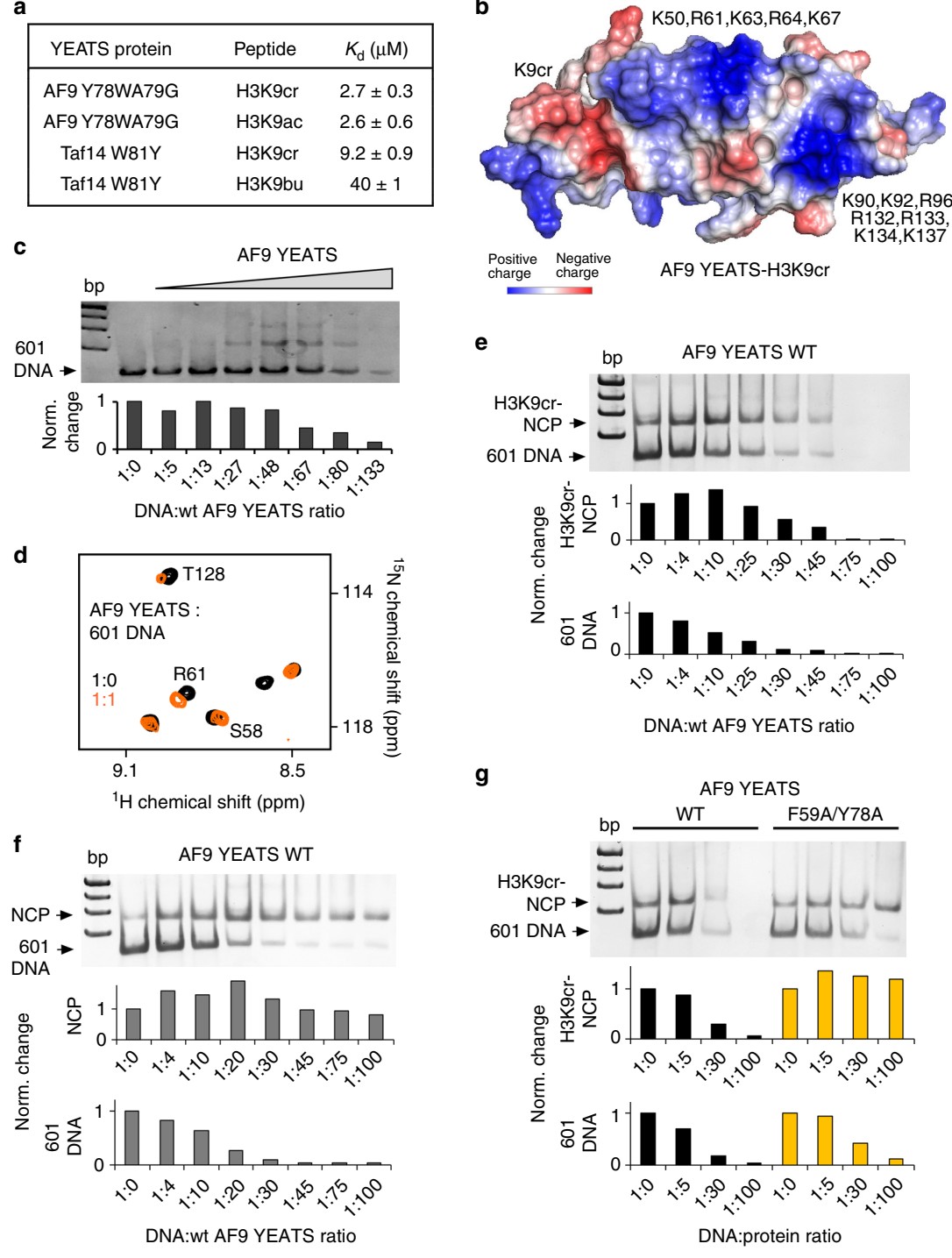

**Fig. 5** AF9-YEATS binds DNA. **a** Binding affinities of AF9-YEATS and Taf14-YEATS to indicated histone peptides. Values represent the average of three separate experiments with error calculated as the SD between the runs. **b** Electrostatic potential surface representation of the AF9-YEATS in complex with H3K9cr generated in PyMol (PDB ID 5hjb). Basic residues are labeled. **c** EMSA with 147-bp 601 DNA (1.88 pmol/lane) incubated with increasing amounts of AF9-YEATS. DNA to protein molar ratio is shown below the gel image. **d** Superimposed $^1$H,$^{15}$N HSQC spectra of uniformly $^{15}$N-labeled AF9-YEATS collected upon titration with 601 DNA. The spectra are color coded according to the protein:DNA molar ratio. **e–g** EMSA assays with 1 pmol/lane H3K9cr-NCP (**e**, **g**) or unmodified NCP (**f**) incubated with increasing amounts of WT AF9-YEATS or F59A/Y78A mutant. Band intensities in (**c**, **e–g**) were quantified by densitometry using ImageJ

discovery of the DNA-binding function for AF9-YEATS expands the family of epigenetic readers capable of bivalently interacting with histone tails and nucleosomal DNA. These include the Tudor domain of PHF1[25,26], the PWWP domains of LEDGF and PSIP1[27,28], the PZP domain of BRPF1[29], and bromodomain of BRDT[30]. It will be interesting in future studies to explore the idea of conservation of DNA-binding activities in YEATS readers derived from different organisms.

## Methods

**Protein expression and purification.** The Taf14 YEATS domain (aa 1–132 and 1–137) in a pGEX-6P1 vector and the AF9-YEATS domain (aa 1–138) in a pET28b

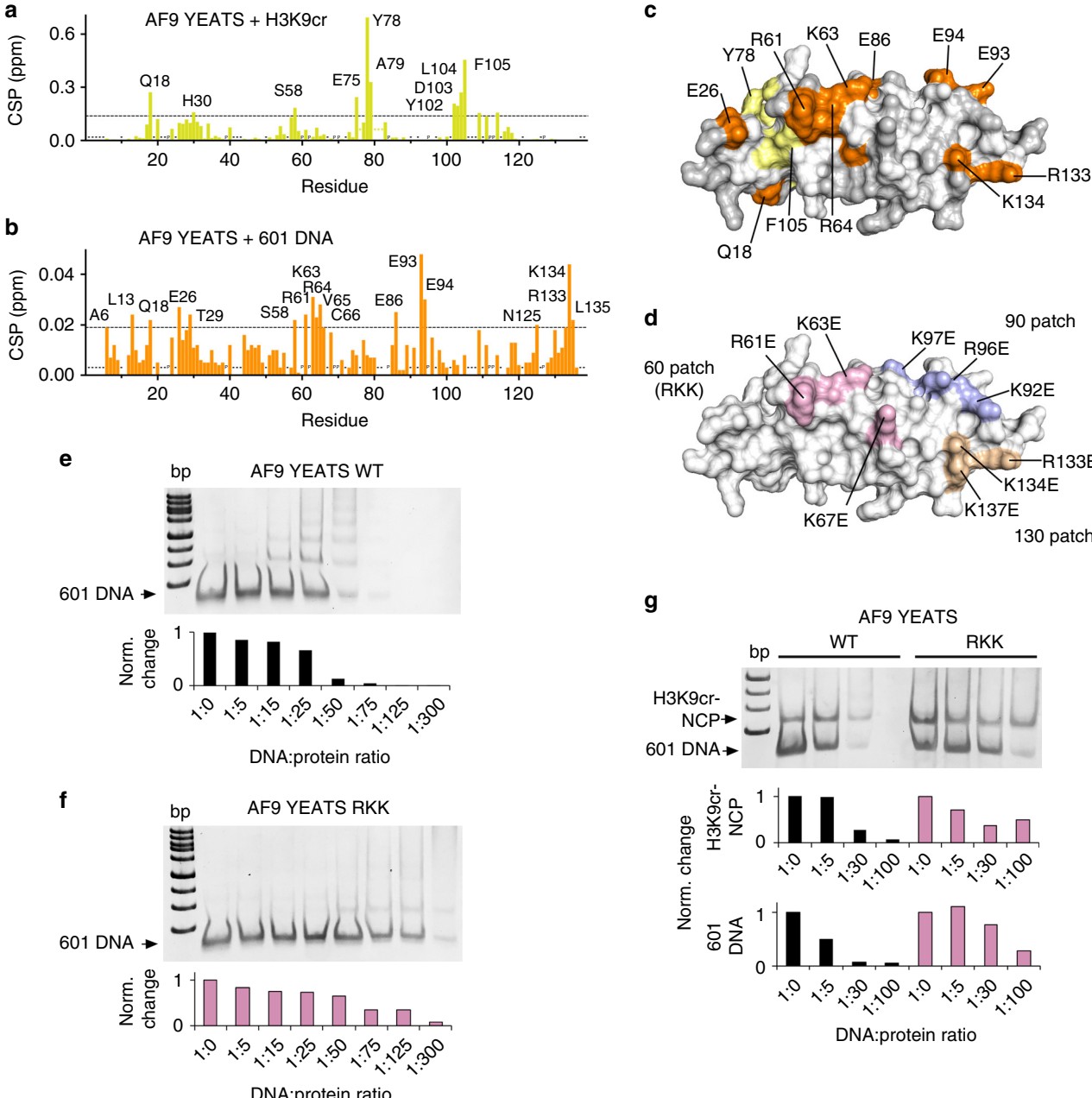

**Fig. 6** DNA- and H3K9cr-binding sites in AF9-YEATS do not overlap. **a**, **b** Analysis of chemical shift perturbations in $^1H,^{15}N$ HSQC spectra of AF9-YEATS caused by (**a**) H3K9cr peptide (1:3 molar ratio) or (**b**) 601 DNA (1:0.5 molar ratio). P indicates a proline residue. '*' indicates that the residue resonances were unassigned in the apo-state. '-' indicates that the residue resonances were unassigned in the H3K9cr-bound state (BMRB 26060). Dashed line indicates selection cut-off (mean + 3 SD). Residues with CSP above the cut-off value are labeled. **c** Identification of the H3K9cr peptide- and 601 DNA-binding sites. Residues with CSP above the cut-off value in (**a**, **b**) are mapped onto the surface of AF9-YEATS (PDB 5hjb), colored yellow and orange, respectively, and labeled. Residues with no assignments or prolines are colored light gray. **d** The 60th patch (R61K/K63E/K67E, light pink), 90th patch (K92E/R96E/K97E, light blue), and 130th patch (R133E/K134E/K137E, wheat) mutations are mapped onto the surface of AF9-YEATS. **e**–**g** EMSA assays with 1 pmol/lane 601 DNA (**e**, **f**) or 1 pmol/lane H3K9cr-NCP (**g**) incubated with increasing amounts of WT AF9-YEATS or RKK mutant. Band intensities were quantified by densitometry using ImageJ

vector were used in this study[13,16]. The YEATS domain mutants of Taf14 (W81Y and G82A) and AF9 (Y78W, Y78W/A79G, R61E/K63E/K67E, K92E/R96E/K97E, R96E/K97E, and R133E/K134E/K137E) were generated using standard Quik-Change site-directed mutagenesis protocols (Stratagene) (Supplementary Figure 12). Wild-type and mutant proteins were expressed in *Escherichia coli* BL21 (DE3) RIL (Agilent) cells grown in either Luria Broth or M9 minimal media supplemented with $^{15}NH_4Cl$ (Sigma-Aldrich). Following induction with isopropyl β-D-1-thiogalactopyranoside (IPTG), cells were harvested by centrifugation and lysed by sonication. In all, 25 mM Tris pH 7.5 buffer, supplemented with 500 mM NaCl, 2 mM β-mercaptoethanol, 1% Triton X-100, and DNase I was used for AF9-

YEATS, and 50 mM HEPES pH 7.5 buffer, supplemented with 150 mM NaCl, 1 mM TCEP, and 0.1% Triton X-100 was used for Taf14-YEATS. GST-fusion proteins were purified on glutathione Sepharose 4B beads (Thermo Fisher), and the GST tag was either cleaved with PreScission protease overnight, or left on and the proteins then were eluted off the resin with 50 mM reduced L-glutathione (Fisher). His tag fusion proteins were purified on a nickel–NTA resin (Qiagen), and the His tag was cleaved with Thrombin. Proteins were further purified by size exclusion chromatography using a S100 column or Hi-Trap SP HP and HiPrep 16/600 Superdex 75 columns (GE Healthcare). The proteins were concentrated in Millipore concentrators and stored at –80 °C.

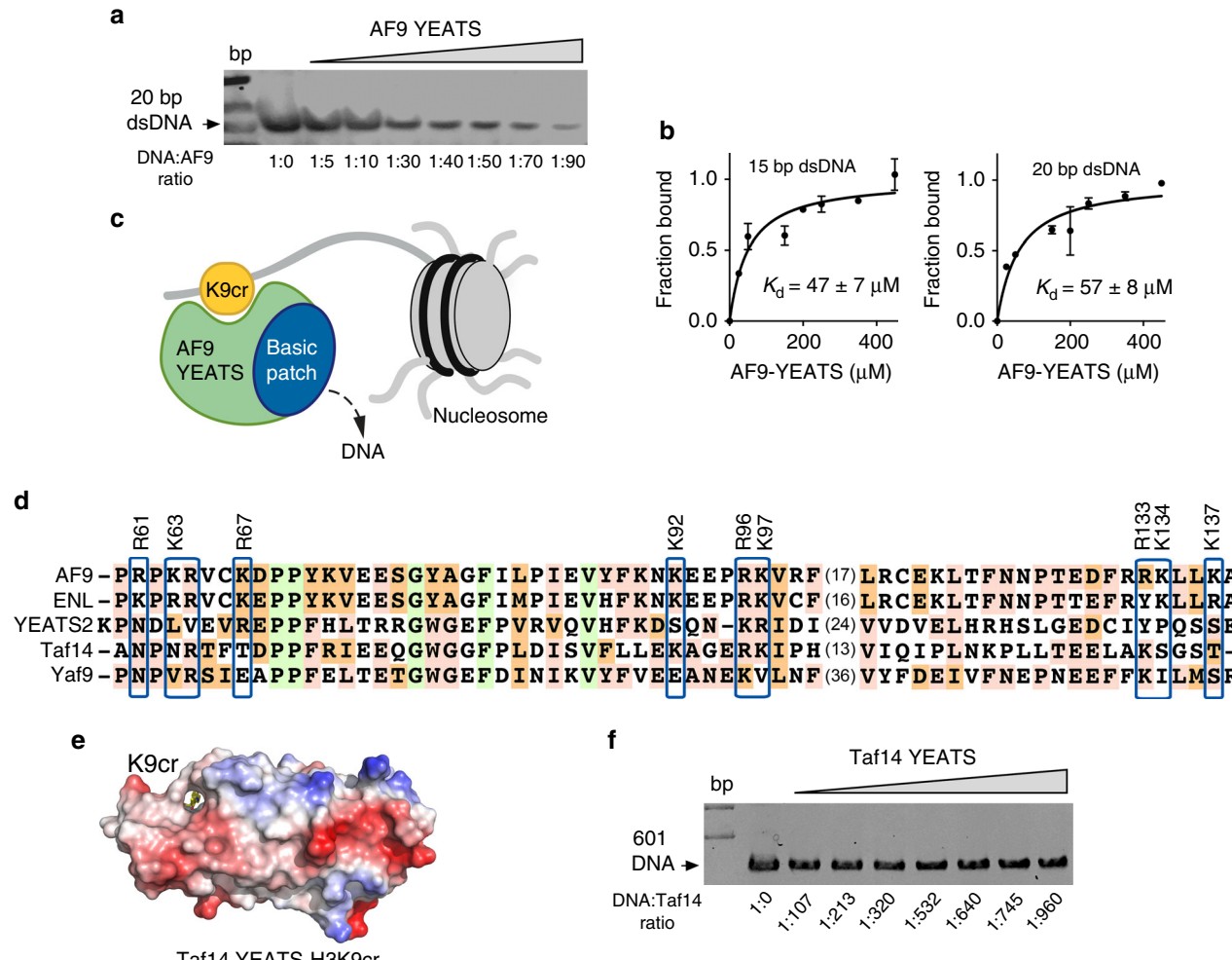

**Fig. 7** Bivalent binding of AF9-YEATS is not conserved in Taf14-YEATS. **a** EMSA with 5 pmol/lane 20 bp dsDNA incubated with increasing amounts of AF9-YEATS. **b** Binding curves used to determine $K_d$ for the DNA:AF9-YEATS complex by EMSA. The band of free DNA was used for quantification of the complex formation. Binding constants are obtained from duplicate measurements as mean ± standard error. **c** A schematic of the bivalent interaction of AF9-YEATS with histone H3K9cr and DNA. **d** Alignment of the YEATS domain sequences: absolutely, moderately, and weakly conserved residues are colored green, orange, and pink, respectively. Positively charged residues in three patches are indicated by blue boxes. The residues of AF9-YEATS mutated in this study are labeled. **e** Electrostatic surface potential of Taf14-YEATS in complex with the H3K9cr peptide. **f** EMSA with 1.88 pmol/lane 601 DNA and increasing concentration of Taf14-YEATS

**X-ray data collection and structure determination**. Taf14-YEATS proteins (1–137) were buffer exchanged and concentrated to 14 mg/mL in 50 mM MES (pH 6.5) at 4 °C. Wild-type Taf14-YEATS was incubated with five molar equivalents of either H3K9bu (5–13) or H3K9pr (5–13) peptide at 25 °C for 30 min prior to crystallization. Crystals were obtained by the sitting-drop vapor diffusion method at 18 °C. The 1.6 µl crystallization drop contained a 1:1 ratio of protein-ligand solution to reservoir solution composed of 44–46% PEG 600 (v/v) and 0.2 M citric acid pH 6.0. Taf14-YEATS G82A was incubated with 10 molar equivalents of the H3K9cr (5–13) and crystallized under the same conditions as wild-type Taf14-YEATS. The mutant in the apo-state was crystallized in a 1:1 ratio with a reservoir solution composed of 48% PEG 600 (v/v) and 0.2 M sodium citrate pH 6.0. X-ray diffraction data were collected at a wavelength of 1.0 or 1.54 Å and a temperature of 100 K on a Rigaku Micromax 007 high-frequency microfocus X-ray generator with a Pilatus 200 K 2D area detector at the UC Denver Biophysical Core facility or on beamline 4.2.2 at the Advance Light Source administrated by the Molecular Biology Consortium.

The AF9-YEATS proteins, Y78W mutant and wild-type, were buffer exchanged and concentrated at 4 °C in 50 mM sodium citrate pH 6.5, 5 mM DTT, or 50 mM Bis-Tris pH 6.5, 5 mM DTT, respectively. AF9-YEATS Y78W was incubated at 6 mg/ml with a two-molar equivalence of H3K9cr (5–14) peptide at RT for 30 min before crystallization. Crystals of AF9-YEATS Y78W in complex with H3K9cr were grown at 4 °C, using the sitting-drop vapor diffusion method against 100 mM MES pH 6.0, 20%(w/v) PEG 6000, and 10 mM ZnCl$_2$ reservoir solution. The 1.0 µl crystallization drop contained a 1:1 ratio of protein-ligand to reservoir solution. Crystals were cryoprotected with 25% (v/v) glycerol. Wild-type AF9-YEATS was

incubated at 6 mg/ml with a two-molar equivalence of H3K9bu (1–19). Crystals were grown at 4 °C, using the sitting-drop vapor diffusion method in a 1:1 ratio of protein-ligand to reservoir solution (200 mM sodium malonate, 100 mM Bis-Tris propane pH 8.5, and 20% (w/v) PEG 3350) in a 1.0 µl crystallization drop and were cryoprotected with 25% (v/v) glycerol. X-ray diffraction data from AF9-YEATS Y78W:H3K9cr and AF9-YEATS:H3K9bu crystals were collected at a wavelength of 1.28 and 0.98 Å, respectively, and a temperature of 100 K on beamline 4.2.2 at the Advance Light Source administrated by the Molecular Biology Consortium.

HKL3000 was used for indexing, scaling, and data reduction[31]. The phase solutions were solved by molecular replacement with Phaser using a modified Taf14 or AF9-YEATS domain as a search model (PDB 5D7E and 4TMP, respectively). Model building was carried out with Coot and refinement was performed with Phenix.Refine[32,33]. Coot and Molprobity were used to verify the model fit to density and model quality[32,34]. The crystallographic and refinement statistics are shown in Supplementary Table 1.

**NMR spectroscopy**. NMR experiments were performed at 298 K on a Varian INOVA 600 MHz spectrometer equipped with a cryogenic probe. $^1$H,$^{15}$N HSQC spectra of 0.1–0.2 mM uniformly $^{15}$N-labeled wild-type or mutated Taf14-YEATS (in phosphate-buffered saline (PBS) buffer pH 6.8, 8% D$_2$O) and AF9-YEATS (in 25 mM Tris-HCl pH 7.5, 250 mM NaCl, 2 mM BME, 8–10% D$_2$O) were recorded in the presence of increasing concentrations of H3K9cr, H3K9bu, H3K9ac, H3K9su, or H3K9hib peptides. $K_d$ value was calculated by a nonlinear least-squares

analysis in Kaleidagraph using the

$$\Delta\delta = \Delta\delta_{max} \frac{\left(([L] + [P] + K_d) - \sqrt{([L] + [P] + K_d)^2 - 4[P][L]}\right)}{2[P]} \quad (1)$$

where $[L]$ is concentration of free acetyllysine, $[P]$ is concentration of the protein, $\Delta\delta$ is the observed chemical shift change, and $\Delta\delta_{max}$ is the normalized chemical shift change at saturation. Normalized chemical shift changes were calculated using the equation

$$\Delta\delta = \sqrt{(\Delta\delta H)^2 + \left(\frac{\Delta\delta N}{5}\right)^2} \quad (2)$$

where $\Delta\delta$ is the change in chemical shift in parts per million (ppm).

**Fluorescence spectroscopy.** Spectra were recorded at 25 °C on a Fluoromax-3 spectrofluorometer (HORIBA). The samples containing 1–2 μM wild-type or mutated YEATS domains in PBS buffer pH 7.4 (Taf14) or 20 mM Tris pH 7.5, 500 mM NaCl, and 2 mM BME (AF9) and progressively increasing concentrations of H3 peptides, H3K9cr, H3K9ac, or H3K9bu (all aa 5–13) (Synpeptide) were excited at 295 or 280 nm. Emission spectra were recorded between 300 and 380 nm with a 1 nm step size and a 0.5 s integration time. $K_d$ values were determined using a nonlinear least-squares analysis and the equation:

$$\Delta I = \Delta I_{max} \frac{\left(([L] + [P] + K_d) - \sqrt{([L] + [P] + K_d)^2 - 4[P][L]}\right)}{2[P]} \quad (3)$$

where $[L]$ is the concentration of the histone peptide, $[P]$ is the concentration of the protein, $\Delta I$ is the observed change of signal intensity, and $\Delta I_{max}$ is the difference in signal intensity of the free and bound states of the protein. $K_d$ values were averaged over three separate experiments, and error was calculated as the standard deviation between the runs.

**Real-time qPCR.** Real-time qPCR was performed essentially as described[35]. Briefly, wild-type or *taf14* mutant yeast cells were cultured in YPD overnight at 30 °C and cell pellets corresponding to 10 OD$_{600}$ equivalents were collected. Total RNA was extracted using the phenol/chloroform method as described[35] and RNA was treated with DNase I to eliminate residual DNA. In all, 2 mg of RNA was used to produce complementary DNA (cDNA) with random hexamer primers and reverse transcriptase IScript (BioRad). cDNA was diluted to 0.1 mg/ml and SYBR green master mix was used for real-time qPCR. *ACT1* was also examined and used as a normalization control. The expression of all Taf14 proteins were equal in abundance, indicating that the expression effects observed are due to Taf14 acyllysine-binding defects. The data shown are from three biological replicates.

**Expression and purification of H3 mutants H3K9ac and H3K9cr.** To incorporate *N*-ε-acetyl-L-lysine (AcK) into the K9 position of histone H3, pETDuet-1 vector encoding human histone H3 with amber stop codon (pETDuet-1-H3K9TAG) introduced at H3K9 was used to co-transform *E. coli* BL21 (DE3)-ΔcobB strain with pEVOL-MmAcKRS. Single colony was picked and inoculated in 2YT medium supplemented with 100 mg/L ampicillin and 34 mg/L chloramphenicol. When OD reached to 0.6, H3 expression was induced at 37 °C by adding 0.5 mM IPTG, 0.2% (w/v) L-arabinose and 5 mM AcK into cell culture. Cells were harvested 6 h after induction, and purified in the same steps as previously reported[36]. The whole procedure of CrK (*N*-ε-crotonyl-L-lysine) incorporation into the K9 position of histone H3 was identical with AcK incorporation, except that pEVOL-MmPylRS-384W vector was used, and 1 mM of CrK (*N*-ε-crotonyl-L-lysine) was added into 2YT medium after induction.

**Assembly of nucleosomes.** Recombinant histone His-TEV-H2A, His-TEV-H2B, and His-SUMO-TEV-H4 were purified in the previously reported steps[36]. All four histone pellets including histone H3K9ac or H3K9cr were dissolved in 6 M GuHCl buffer (6 M guanidinium hydrochloride, 20 mM Tris, 500 mM NaCl, pH 7.5), and concentration was measured by ultraviolet (UV) absorption at 280 nm (Biotek synergy H1 plate reader). To prepare H2A/H2B dimer, His-TEV-H2A and His-TEV-H2B were mixed in the molar ratio of 1:1, and 6 M GuHCl buffer was added to adjust total protein concentration to 4 μg/μl. Denatured His-TEV-H2A/His-TEV-H2B solution was dialyzed sequentially at 4 °C against 2 M TE buffer (2 M NaCl, 20 mM Tris, 1 mM EDTA, pH 7.8), 1 M TE buffer (1 M NaCl, 20 mM Tris, 1 mM EDTA), 0.5 M TE buffer (0.5 M NaCl, 20 mM Tris, 1 mM EDTA). Then, the resulting dimer solution was centrifuged for 5 min at 4 °C to remove precipitates, and the concentration of dimer was determined by UV absorption at 280 nm. Steps of His-TEV-H3/His-SUMO-TEV-H4 tetramer refolding were generally the same as His-TEV-H2A/His-TEV-H2B dimers except that the total protein concentration should be adjusted to 2 μg/μl and no stirring in the sequential dialysis. Then His-TEV-H2A/Hi-TEV-H2B dimers were mixed with His-TEV-H3/His-SUMO-TEV-H4 tetramers in a molar ratio of 1:1 to generate histone octamers, and NaCl solid was added to adjust

NaCl concentration to 2 M. The 147-bp biotinylated 601 nucleosome positioning was prepared by polymerase chain reaction with biotinylated primers and purified with PCR cleanup kit (#2360250 Epoch Life Science). Purified 147-bp DNA was re-dissolved in 2 M TE buffer and added to histone octamer solution in the molar ratio of 0.85:1. In all, 2 M TE buffer was added to adjust final 147 bp DNA concentration to 2–3 μM. The DNA histone mixture solution was then transferred to a dialysis bag and placed inside about 200 ml 2 M TE buffer, while stirring at room temperature, Tris buffer with no salt (20 mM Tris) was slowly added into the 2 M salt buffer through a liquid transfer pump (#23609-170 VWR®). Nucleosomes formed when salt concentration was reduced to around 150 mM (measured by EX170 salinity meter), and the DNA histone mixture solution was further dialyzed into low salt Tris buffer (20 mM Tris, 20 mM NaCl, 0.5 mM EDTA, pH 7.8). Precipitates were removed by centrifuge, and the concentration of the nucleosomes was measured by A260 reading using the Biotek synergy H1 plate reader. His-TEV protease was added to nucleosome solutions (TEV:nucleosome 1:30, w-w) to remove all the histone tags after incubation for 1 h at 37 °C, and finally all the His tagged impurities were removed from nucleosome solution by Ni$^{2+}$-NTA resin (Thermo Fisher #88221).

**Peptide pulldown assays.** One microgram of biotinylated histone peptides with different modifications were incubated with 1–2 μg of GST-fused proteins in binding buffer (50 mM Tris-HCl 7.5, 150 mM NaCl, 0.1% NP-40, 1 mM phenylmethylsulfonyl fluoride (PMSF)) overnight. Streptavidin beads (Amersham) were added to the mixture, and the mixture was incubated for 1 h with rotation. The beads were then washed three times and analyzed using sodium dodecyl sulfate–polyacrylamide gel electrophoresis (SDS–PAGE) and western blotting. Anti-GST (sc-459, 1:1000) antibody was from Santa Cruz (Supplementary Figure 13).

**MOs computations.** All computations were performed on a Linux HPC cluster with the Gaussian 09 quantum chemical software package (Gaussian 09, Revision A.02, M. J. Frisch et al., Gaussian, Inc., Wallingford CT, 2016). X-ray coordinates of the carbon, oxygen, and nitrogen atoms belonging to the triple-stacked truncated trp–crotonamide–phe fragment were frozen, whereas the positions of the attached hydrogen atoms were fully optimized at the B3LYP/6-311+G(d,p) level of DFT theory producing the truncated structure (Supplementary Figure 6). XYZ coordinates for the truncated triple-stack structure with fully optimized hydrogens are available upon request.

**EMSA.** A total of 32 repeats of the 601 Widom DNA sequence were cloned into the pJ201 plasmid and transformed into DH5α cells. The plasmid was purified as previously described[25] and by Qiagen-QIAprep Spin Miniprep kit. Separation of the individual sequences was completed by digestion of the plasmid with EcoRV. The 601 Widom DNA was purified from the remaining plasmid by gel extraction (Qiagen-MinElute Gel Extraction kit). Annealed DNA was made up of single-stranded oligos (IDT) in water brought up to 95 °C for 20 min then cooled to 16 at 0.1 °C/s. Increasing amounts of the Taf14 and AF9-YEATS domains were incubated with 601 Widom DNA (1–3 pmol/lane), annealed double-stranded DNA in buffer (20 mM Tris-HCl pH 7.5 and 150 mM NaCl), or nucleosome in buffer (20 mM Tris-HCl pH 7.5,150 mM NaCl, 0.1 mM EDTA, and 10% glycerol) for 30 min at room temperature. Specifically, for each lane, 1.88 pmol (Figs. 5c, 7f) or 1 pmol (Figs. 6e, f) 601 DNA, 5 pmol 20-mer DNA (Figs. 7a), or 5 pmol 15-mer DNA (Supplementary Figure 11a) was incubated with increasing amounts of mutant or wild-type AF9-YEATS or Taf14-YEATS. In all, 1 pmol/lane (Figs. 5e-g and 6g and Supplementary Figure 10c) of nucleosome was incubated with increasing wild-type or mutant AF9-YEATS domain. The reaction mixtures were loaded on 5–10% native polyacrylamide gels and electrophoresis was performed in 0.2 × TB or TBE buffer (1 × TB/TBE = 90 mM Tris, 64.6 mM boric acid, with or without 2 mM EDTA) at 100–130 V on ice. Gels were stained with ethidium bromide or SYBR Gold (Thermo Fisher) and visualized at 365 mm or by Blue LED (UltraThin LED Illuminator- GelCompany). Quantification of gel bands was performed using ImageJ. Each EMSA experiment was performed five times (Figs. 5c, 6e), four times (Figs. 7a, f and Supplementary Figure 11a), three times (Fig. 5e), twice (Fig. 5f and Supplementary Figure 10c), and once (Figs. 5g and 6f, g).

## Data availability
The atomic coordinates and structure factors of Taf14-YEATS and AF9-YEATS have been deposited in the Protein Data Bank under the accession codes 6MIQ, 6MIP, 6MIO, 6MIN, 6MIM, and 6MIL. The data that support this work are available from the corresponding author upon reasonable request.

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

## Acknowledgements

We thank Jinyong Zhang, Ruben Rosas-Ospina, and Jayce Breig for help with experiments, Jay Nix at beamline 4.2.2 of the ALS in Berkeley for help with X-ray crystallographic data collection, and Ming-Ming Zhou for a kind gift of cDNA of the AF9-YEATS F59A/Y78A mutant. This work was supported in part by grants from NIH GM106416, GM100907, and GM125195 to T.G.K., GM121584 to W.R.L., R35GM126900 to B.D.S., and CA204020 to X.S., and from NSF CHE-1665342 to A.G.K. X.S. is a Leukemia & Lymphoma Society Career Development Program Scholar (1339-17).

## Author contributions

B.J.K., K.R.V., F.H.A., W.W.W., J.Z., Y.Z., A.A.B., W.M., and Y.L. performed experiments and together with H.L., X.S., A.G.K., B.D.S., W.R.L., and T.G.K. analyzed the data. B.J.K., K.R.V., F.H.A., and T.G.K. wrote the manuscript with input from all authors.

## Additional information

**Competing interests:** The authors declare no competing interests.

