## [peer review file · Nature Communications]

Reviewer #1 (Remarks to the Author):

It is now evident that histone modifications play a crucial role in epigenetic regulations. Recent studies have shown a variety of chemical modifications on histones that are previously unrecognized, including lysine crotonylation, which has been reported to stimulate gene expression. Kutateladze and co-workers have previously demonstrated through structural analysis that the YEATS domains of yeast Taf14 and human AF9 function as a reader of histone H3 crotonylation. In this study, Klein et al. conducted a further detailed investigation on the recognition mechanism of histone crotonylation by the YEATS domains, including mutational analysis on key amino acids within the domain. In addition, the authors show that human, but not yeast, YEATS domains may possess a DNA-binding activity. Thus, this study may provide insights into the YEATS-histone interactions potentially important for epigenetic regulations.

Although extensive structural analyses have been carried out in this study, which are of high quality, the main part of the study (Figs. 1-3) seems confirmatory follow-ups of the previous studies (e.g., Andrews et al. 2016 Nat Chem Biol, Li et al. 2016 Mol Cell), rather than addressing novel regulatory mechanisms. In contrast, the potential DNA-binding activity of the human YEATS domains may be of interest (Fig. 4).

In this study, however, only limited biochemical results, using free DNA fragments, are provided. The interaction between the basic patch of the AF9 YEATS and DNA has been suggested to be electrostatically driven (Fig. 4d), and the domains could bind to nucleosomal DNA, as illustrated in Fig. 4g. However, the charge of nucleosomal DNA would be neutralized by the strong basicity of histones, and the DNA wrapping histones might be structurally inaccessible. Thus, additional biochemical experiments using nucleosomes may be needed, e.g., as performed in Fig. 1e.

Additionally or alternatively, one can perform cell-based assays, such as biochemical fractionation and ChIP(-seq). For these experiments, designing mutant proteins with less DNA binding may be needed. I think that without further biochemical analyses, the potential role of the DNA-binding activity, if any, remains elusive.

Overall, the study has a potential to provide a clue for a novel epigenetic mechanism but is currently lacking the supporting biochemical evidence. Given the high standard of the journal, I feel that this manuscript as a current form would not satisfy the requirement for publication.

Reviewer #2 (Remarks to the Author):

This study used structure-based analysis to understand the selectivity of two YEATS domains, Taf14 and AF9. Taf14 binds to crotonyllysine with 3 fold higher affinity than saturated acyllysines and the authors concluded that pi stacking with both crotonyl amide and crotonyl alkene plays a critical role in such higher affinity binding. AF9 has no selectivity against saturated vs unsaturated acyllysines because the domain only engages with the modified lys via pi stacking with acyl mide. AF9 is capable of binding to DNA whereas Taf14 does not. These studies are well-characterized and interesting, which should impact the field significantly. I only have three minor questions:

1. Is the 3-fold affinity enhancement significant? It does not seem to be a dramatic enhancement. I feel the authors may want to discuss this a little bit to emphasize the biological importance of such enhancement.
2. In Fig 3d, where are blue peaks? I can only see black (1:0) and yellow (1:5) peaks but not blue (1:3) peaks.
3. Is the format of the manuscript for Nat Commun? Significant rewriting appears necessary to comply with the journal format.

Reviewer #3 (Remarks to the Author):

The YEATS domain recently emerged as a novel epigenetic reader with strong disease association. YEATS domains can be recruited to diverse and varying acyl marks. Structural determinants for these selectivity profiles, which differentiate YEATS from BROMO domains, are poorly understood, and are the primary and timely focus of this manuscript.

The authors first conduct an in-depth structural analysis supported by site-directed mutagenesis to analyze the structural mechanisms underlying the different selectivity of YEATS domains found in yTaf14 and hAF9 for acetylated, butyrylated and crotonylated histones. The analysis is thorough and compelling, though a few open questions should be addressed in this reviewer's opinion. The authors then reveal a DNA-binding activity of human AF9, probably shared by other human YEATS domain, which suggests a bivalent mode of binding to chromatin relying on both DNA and acyl-lysine recognition but is not further investigated.

This manuscript reveals structural features of YEATS domains that are relevant to their function and to the design of chemical inhibitors. As such, it deserves publication in this Journal, but a few points should be addressed first.

Fig 1b. Even though it was previously published, it would be useful to include K_d values for the acetylated peptide in the same assay, for comparison with H3K9cr and H3K9bu values. It would also make the comparison more consistent with Fig 1d where band intensities can be compared between the three marks in the context of a pull-down assay with histone peptides.

Fig 1e: The pull-down assay with reconstituted nucleosome shown here is biologically more relevant, but the data for H3K9bu is currently missing and should be added.

I.128: W82 should read W81.

I.170: The conserved selectivity of Taf14 W81Y for H3K9cr ($K_d=9.2 \mu\text{M}$) versus H3K9ac ($K_d=40 \mu\text{M}$) does not seem to agree with the overall narrative and should be explained.

I.181: Could the authors explain how they estimated interaction of AF9 with 14 major/minor DNA grooves?

I. 182: this reviewer sees no significant difference in the EMSA assay conducted at 150 and 250 mM salt concentration (Fig. 4d): Fig 4d does not show that binding to DNA is driven by electrostatic interactions. Also, DNA:AF9 ratios are missing from Fig 4d.

fig 4e: it is surprising that, unlike what is seen in figs 4c,d, no shifted band corresponding to AF9-DNA complex appear with increasing concentrations of AF9, as the band corresponding to unbound DNA disappears.

The manuscript is composed of two distinct sections: binding of YEATS domains to acyl-lysine and binding of human AF9 YEATS domain to DNA. It would be good to show how mutations that affect acyl-lysine recognition and mutations that affect DNA binding (candidates include K67A or K134A)

differentially decrease the affinity of AF9 YEATS to nucleosome. It would indicate the relative contributions of the acyllysine- and DNA-binding events to chromatin association. This would bring the two independent sections of the manuscript into a unified model and raise the profile of this manuscript.

Reviewer #4 (Remarks to the Author):

YEATS domain's are an exciting class of effector domains for recognizing acylated histones, and their molecular recognition with crotonylated lysine, versus acetylated lysine complements studies with bromodomains and provides useful insight into epigenetic regulation. The authors of this report provide detailed structural and quantitative information regarding YEATS domain acylated lysine recognition, via x-ray crystallography, NMR, and pull-down experiments. Further, they seek to rationalize the acyl lysine recognition via aromatic interactions utilizing a WG motif which restricts a favorable tryptophan rotamer population. In the final part of the manuscript they briefly analyze an additional electrostatic interaction of the AF9-YEATS domain with DNA. Although the data was clearly presented and the conclusions are supported by the data, the questions being asked could be further flushed out with more experimental data. Nonspecific interactions with DNA are becoming more prevalently observed with epigenetic effector domains, however in this case, the level of structural detail for the interaction is still lacking. In addition the level of selectivity gained in the chimeric YEATS domain, seems to be less useful given the > 10 fold reduction of affinity. At this point for publication in this journal, although the findings are interesting and important further analysis is suggested. Detailed suggestions are listed below.

1. The authors carry out recognition experiments with modified nucleosome core particles with defined modifications, which is commendable. However, if one of the key points was to show the selectivity differences for acylated lysine between TAF14-YEATS and AF9-YEATS, it was unclear why the AF9-YEATS interaction was not tested as well. Due to their data supporting DNA interactions, a control with the unmodified histone in the NCP, would also be useful to include. Similar to figure 1D. This experiment would complement their DNA interactions studies at the end of the report.

2. The authors make a mutant protein in the TAF14-YEATS for reducing binding to acetylated Lysine. The first challenge with this analysis was that as written it is hard to appreciate how much selectivity has been engineered. Butrylated lysine was shown to bind ~3 fold weaker than crotonylated, in Figure 1B, but acetylated lysine was not included in this table. Therefore from the data it is unclear how much of an improvement was made.

2b. Although the binding to crotonylated lysine was reduced from 9 micromolar to 124 micromolar, the authors rationalize this was in the range of affinities of bromodomain interactions, supporting physiological relevance. A challenge from that comparison, is that many bromodomain-containing proteins have additional effector domains for increasing affinity, such as a PHD domain. A pull-down experiments might be useful to demonstrate if this is a reasonable affinity.

3. On line 189 page 8, the authors use their EMSA data with double stranded DNA to support a bivalent interaction with both histones and DNA in the nucleosome. However, at this point the authors have not shown if the interactions are antagonistic or complementary to one another, or structurally how the interactions are occurring. NMR or ITC might be useful to show if a ternary complex is formed.

3b. There are several HSQC experiments that are presented. Assigning one of the spectra for the YEATs domains would help structurally characterize the interaction further beyond a basic patch. The data at this point cannot pinpoint if this is a specific or non-specific effect. Citing other reports of DNA effector domain interactions would be relevant for this report as well. E.g see work of Miller et al. Nat. Commun. 2016. On BET bromdomain-DNA interactions.

Minor: In the supporting information and the main text, the binding isotherms presented for the intrinsic fluorescence measurements show multiple curves, but the symbols are not identified.

We thank the Editor and Reviewers for the insightful and very constructive comments, which were helpful in revising and strengthening this manuscript.

Reviewer 1, Comment 1: In this study, however, only limited biochemical results, using free DNA fragments, are provided. The interaction between the basic patch of the AF9 YEATS and DNA has been suggested to be electrostatically driven (Fig. 4d), and the domains could bind to nucleosomal DNA, as illustrated in Fig. 4g. However, the charge of nucleosomal DNA would be neutralized by the strong basicity of histones, and the DNA wrapping histones might be structurally inaccessible. Thus, additional biochemical experiments using nucleosomes may be needed... Additionally or alternatively, one can perform cell-based assays... For these experiments, designing mutant proteins with less DNA binding may be needed. I think that without further biochemical analyses, the potential role of the DNA-binding activity, if any, remains elusive.

Author's response: As suggested, we have thoroughly investigated binding of AF9 YEATS to nucleosomes and DNA using EMSA, NMR and mutagenesis. The new results, shown in Figs. 5d-g, 6a-g, 7d, Suppl. Fig. S10, confirmed the interaction with the nucleosomes. In addition, we have mapped the AF9:DNA binding interface using NMR experiments, mutated the DNA-binding site residues, and showed that the H3K9cr-binding site and the DNA-binding site do not overlap (Fig. 6a-c).

Reviewer 2, Comment 1: I only have three minor questions:

1. Is the 3-fold affinity enhancement significant? It does not seem to be a dramatic enhancement. I feel the authors may want to discuss this a little bit to emphasize the biological importance of such enhancement. – **as suggested, we have included the following sentence on page 16: The enhancement in selectivity of Taf14-YEATS to crotonyllysine is comparable to the enhancement in selectivity of other well-recognized epigenetic readers, such as DPFs.^{8,11}**
2. In Fig 3d, where are blue peaks? I can only see black (1:0) and yellow (1:5) peaks but not blue (1:3) peaks. – **blue peaks are underneath yellow peaks (please see a zoom-in image on the right).**
3. Is the format of the manuscript for Nat Commun? Significant rewriting appears necessary to comply with the journal format. – **we have reformatted the manuscript accordingly.**

Reviewer 3, Comment 1: Fig 1b. Even though it was previously published, it would be useful to include Kd values for the acetylated peptide in the same assay, for comparison with H3K9cr and H3K9bu values. It would also make the comparison more consistent with Fig 1d where band intensities can be compared between the three marks in the context of a pull-down assay with histone peptides.

Author's response: we agree, we have measured binding affinity of Taf14 YEATS to H3K9ac (Suppl. Fig. S1, right panel) and added the value in Fig. 1b.

Reviewer 3, Comment 2: Fig 1e: The pull-down assay with reconstituted nucleosome shown here is biologically more relevant, but the data for H3K9bu is currently missing and should be added.

Author's response: unfortunately, H3K9bu-nucleosomes or histone H3K9bu (for reconstitution) are not available to us, and we could not perform this assay.

Reviewer 3, Comment 3: l.128: W82 should read W81. – **we have corrected this typo, thank you.**

Reviewer 3, Comment 4: I.170: The conserved selectivity of Taf14 W81Y for H3K9cr (Kd=9.2 μM) versus H3K9ac (Kd=40 μM) does not seem to agree with the overall narrative and should be explained.

Author's response: we have included an explanation on page 9.

Reviewer 3, Comment 5: I.181: Could the authors explain how they estimated interaction of AF9 with 14 major/minor DNA grooves?

Author's response: we have added the citation #23 and revised this phrase to ... multiple major/minor grooves²³ ... (page 9).

Reviewer 3, Comment 6: I. 182: this reviewer sees no significant difference in the EMSA assay conducted at 150 and 250 mM salt concentration (Fig. 4d): Fig 4d does not show that binding to DNA is driven by electrostatic interactions. Also, DNA:AF9 ratios are missing from Fig 4d.

Author's response: we agree, the salt concentration assays have been removed, and new EMSA data are now included (Figs. 5e-g and 6e-g).

Reviewer 3, Comment 7: fig 4e: it is surprising that, unlike what is seen in figs 4c,d, no shifted band corresponding to AF9-DNA complex appear with increasing concentrations of AF9, as the band corresponding to unbound DNA disappears.

Author's response: the shift band is often difficult to see for 15bp and 20bp DNA:AF9-YEATS complexes and it is more obvious on 10% gel (please see Fig. on the left that shows EMSA with 20 bp DNA and progressively increasing concentrations of AF9).

Reviewer 3, Comment 8: It would be good to show how mutations that affect acyllysine recognition and mutations that affect DNA binding (candidates include K67A or

K134A) differentially decrease the affinity of AF9 YEATS to nucleosome. It would indicate the relative contributions of the acyllysine- and DNA-binding events to chromatin association. This would bring the two independent sections of the

manuscript into a unified model and raise the profile of this manuscript.

Author's response: as suggested, we explored F59A/Y78A (the mutant defective in H3K9cr binding) and R61E/K63E/K67E (the mutant defective in DNA binding) in EMSA assays. The new data are shown in Figs. 5e-g and 6e-g and described on pages 10-11.

We have generated other mutants of AF9 YEATS, including K92E/R96E/K97E, R96/K97E, and R133E/K134E/K137E, however they were insoluble (please see Fig. above and data not shown).

Reviewer 4, Comment 1: The authors carry out recognition experiments with modified nucleosome core particles with defined modifications, which is commendable. However, if one of the key points was to show the selectivity differences for acylated lysine between TAF14-YEATS and AF9-YEATS, it was unclear why the AF9-YEATS interaction was not tested as well. Due to their data supporting DNA interactions, a control with the unmodified histone in the NCP, would also be useful to include. Similar to figure 1D. This experiment would complement their DNA interactions studies at the end of the report.

Author's response: as suggested, we have thoroughly investigated binding of AF9 YEATS to nucleosomes and DNA using EMSA, NMR and mutagenesis. The new results are shown in Figs. 5d-g, 6a-g, 7d, Suppl. Fig. S10. In addition, we have mapped the AF9:DNA binding interface using NMR experiments, mutated the DNA-binding site residues, and showed that the H3K9cr-binding site and the DNA-binding site do not overlap (Fig. 6a-c).

Reviewer 4, Comment 2: The authors make a mutant protein in the TAF14-YEATS for reducing binding to acetylated Lysine. The first challenge with this analysis was that as written it is hard to appreciate how much selectivity has been engineered. Butyrylated lysine was shown to bind ~3 fold weaker than crotonylated, in Figure 1B, but acetylated lysine was not included in this table. Therefore from the data it is unclear how much of an improvement was made.

Author's response: we agree, we have measured binding affinity of Taf14 YEATS to H3K9ac (Suppl. Fig. S1, right panel) and added the value in Fig. 1b.

Reviewer 4, Comment 3: Although the binding to crotonylated lysine was reduced from 9 micromolar to 124 micromolar, the authors rationalize this was in the range of affinities of bromodomain interactions, supporting physiological relevance. A challenge from that comparison, is that many bromodomain-containing proteins have additional effector domains for increasing affinity, such as a PHD domain. A pull-down experiments might be useful to demonstrate if this is a reasonable affinity.

Author's response: we have tested the Taf14-YEATS G82A mutant in yeast cells and found that it indeed affects Taf14 target gene transcripts differently compared to the effects of either the *taf14* W81A mutant, *TAF14* wild type, or the *TAF14* deletion. New data are shown in Fig. 2e and discussed on page 6.

Reviewer 4, Comment 4: On line 189 page 8, the authors use their EMSA data with double stranded DNA to support a bivalent interaction with both histones and DNA in the nucleosome. However, at this point the authors have not shown if the interactions are antagonistic or complementary to one another, or structurally how the interactions are occurring. NMR or ITC might be useful to show if a ternary complex is formed.

3b. There are several HSQC experiments that are presented. Assigning one of the spectra for the YEATS domains would help structurally characterize the interaction further beyond a basic patch. The data at this point cannot pinpoint if this is a specific or non-specific effect. Citing other reports of DNA effector domain

interactions would be relevant for this report as well. E.g see work of Miller et al. Nat. Commun. 2016. On BET bromdomain-DNA interactions.

Author's response: as suggested, we have mapped the AF9:DNA binding interface using NMR experiments, mutated the DNA-binding site residues, and showed that the H3K9cr-binding site does not overlap with the DNA-binding site (Fig. 6a-c). We have also shown that both interactions, with H3K9cr and DNA are necessary, as binding of AF9 YEATS F59A/Y78A (the mutant defective in H3K9cr binding) and R61E/K63E/K67E (the mutant defective in DNA binding) is substantially decreased in EMSA assays. The new data are shown in Figs. 5e-g and 6e-g and described on pages 10-11.

We now cite the Miller et al 2016 paper, thank you for pointing to this study; we have also added citations re other readers, such as PZP, PWWP, and Tudor that also bivalently associate with histone tails and DNA (page 12).

Reviewer 4, Comment 5: Minor: In the supporting information and the main text, the binding isotherms presented for the intrinsic fluorescence measurements show multiple curves, but the symbols are not identified.

Author's response: the multiple curves in Fig. 2d and Suppl. Fig. S1 are triplicate experiments that were used to calculate K_d values.

Reviewer #1 (Remarks to the Author):

I think the authors have extensively addressed the points raised in the peer review process; especially the DNA binding activity of AF9-YEATS on nucleosomes has been experimentally characterized by NMR, mutational, and biochemical studies (figs. 5 and 6). However, I was wondering how many times EMSA and quantification was independently repeated (fig. 5e-g, 6e-g). In addition, it might be helpful for readers if the results regarding the point above are more carefully described and discussed (page 10, lines 244-247). Otherwise I feel the manuscript is now suitable for publication.

Reviewer #2 (Remarks to the Author):

I am satisfied with the response to my questions and the revision

Reviewer #3 (Remarks to the Author):

The authors adequately addressed the points raised by this reviewer. I recommend publication of the revised manuscript in this Journal.

Reviewer #4 (Remarks to the Author):

In the authors' revised manuscript, they structurally characterize two different YEATS domains and subsequent mutants on their ability to selectively recognize different acylated histone modifications. They further test an additional interaction with DNA. Several points are raised below for further clarification on their experiments with both histone and DNA interactions. From the experiments presented, the mechanistic cartoon in Figure 7 does not seem to be entirely supported by the data yet.

In the context of the Taf14-YEATS histone binding interactions. The authors claim to have engineered a selective crotonylated lysine reader over acetylation via a G82A mutation. From the

data presented, it is unclear if an increased selectivity has actually been engineered or the affinity just weakened across the board. The starting selectivity was already ~14 fold favoring crotonylation. In their G82A mutant they destabilized binding to crotonylated lysine ~14 fold yielding a 130 micromolar Kd. If the acetylated lysine was similarly weakened it would have a Kd around 2 mM. The intrinsic fluorescence study may be able to quantify this.

I would disagree with the authors statement that association was almost negligible based on small chemical shifts. However at the 10 eq. there does indeed seem to be significant chemical shifts consistent with an expected low millimolar binder.

The yeast gene expression study was a nice indirect way to assess removing acetylation effects on transcription. Their results are consistent with their claims, but could the authors rule out differences in protein expression between the mutants as an alternative to removing acetylated histone binding effects?

On Figure 5C, I would add densitometry analysis. It is hard to see a decrease in the 601 Widom DNA intensity with a corresponding increase of the upper supershifted band. Both bands seem to decrease in intensity at higher concentrations

In subsequent EMSA figures, the authors should add a comment in the main text why only a disappearance of the bands are observed with an absence of a higher supershifted band for the complex.

In Figure 5D in the HSQC experiments, the author mention a gradual addition of DNA to the protein, but only a 1:1 condition is shown. Showing the other concentrations would give a feel for dose dependent effects. The magnitude of chemical shift change in Figure 6 c look quite small.

The authors' YEATS mutant studies and NCP study with unmodified histones versus free DNA support free DNA binding but seem to suggest that the bivalent nucleosomal DNA interaction and histone interaction is not as relevant as the cartoon in Figure 7 would suggest. The EMSA effects with the NCP with crotonylated lysine could also equally be explained by a monovalent interaction.

Experimental. It was very difficult to evaluate conditions used in the experiments. If molar ratios are provided, I would suggest including the protein or DNA concentration in the figure legend rather than putting general conditions in the experimental section.

We thank the Editor and Reviewers for the insightful and very constructive comments, which were helpful in revising and strengthening this manuscript.

Reviewer 1, Comment 1: I think the authors have extensively addressed the points raised in the peer review process; especially the DNA binding activity of AF9-YEATS on nucleosomes has been experimentally characterized by NMR, mutational, and biochemical studies (figs. 5 and 6). However, I was wondering how many times EMSA and quantification was independently repeated (fig. 5e-g, 6e-g). In addition, it might be helpful for readers if the results regarding the point above are more carefully described and discussed (page 10, lines 244-247). Otherwise I feel the manuscript is now suitable for publication.

Author's response: as suggested, we have added the following sentence to the EMSA methods section, page 30: Each EMSA experiment was performed five times (Figs. 5c and 6e), four times (Figs. 7a, 7f and S11a), three times (Fig. 5e), twice (Fig. 5f and S10b), and once (Figs. 5g, 6f and 6g).

The EMSA experiment is described in detail on page 9 (first paragraph).

Reviewer 4, Comment 1: The authors claim to have engineered a selective crotonylated lysine reader over acetylation via a G82A mutation. From the data presented, it is unclear if an increased selectivity has actually been engineered or the affinity just weakened across the board. The starting selectivity was already ~14 fold favoring crotonylation. In their G82A mutant they destabilized binding to crotonylated lysine ~14 fold yielding a 130 micromolar Kd. If the acetylated lysine was similarly weakened it would have a Kd around 2 mM. I would disagree with the authors statement that association was almost negligible based on small chemical shifts. However at the 10 eq. there does indeed seem to be significant chemical shifts consistent with an expected low millimolar binder.

Author's response: we have analyzed NMR titration data for G82A by plotting H3K9ac-induced CSPs vs. H3K9ac concentration. The data fitting yielded a straight line (please see Figure on the left). Because NMR measurements can detect very weak interactions (typically up to ~10 mM), these results indicate that the G82A mutant's affinity toward H3K9ac is lower than 10 mM, and therefore its selectivity for H3K9cr is greater than 80-fold. Thus, these data are in full agreement with the idea of the substantial contribution of the pi-pi-pi interaction.

Reviewer 4, Comment 2: The yeast gene expression study was a nice indirect way to assess removing acetylation effects on transcription. Their results are consistent with their claims, but could the authors rule out differences in protein expression between the mutants as an alternative to removing acetylated histone binding effects?

Author's response: we have added the following sentence to the real Time-qPCR method section, page 26: The expression of all Taf14 proteins were equal in abundance, indicating that the expression effects observed are due to Taf14 acyl-lysine binding defects.

Reviewer 4, Comment 3: On Figure 5C, I would add densitometry analysis. It is hard to see a decrease in the 601 Widom DNA intensity with a corresponding increase of the upper supershifted band. Both bands seem to decrease in intensity at higher concentrations. In subsequent EMSA figures, the authors

AF9-YEATSEMSA of K9cr nucleosome with increasing AF9-YEATS
WT cleaved protein

SYBR GOLD

Protein	Nuc only	nuc:AF9	nuc:AF9	nuc:AF9	nuc:AF9	nuc:AF9	nuc:AF9	601 only
Ratio	1:0	1:2.5	1:5	1:10	1:20	1:30	1:45	1:0

Buffer final concentrations: 20mM Tris-HCl 7.5t, 150mM NaCl,
0.1mM EDTA, 12% glycerolIncubated 30 min at RT before
running on gel5% Acry,
TBE gel
07.19.2018

should add a comment in the main text why only a disappearance of the bands are observed with an absence of a higher supershifted band for the complex.

Author's response: as suggested, densitometry analysis has been added in Fig. 5c. The shift bands are often difficult to see for NCP:AF9-YEATS complexes, and they are more obvious when SYBR GOLD is used for staining, but they are present (please see Figure on the left).

Reviewer 4, Comment 4: In Figure 5D in the HSQC experiments, the author mention a gradual addition of DNA to

the protein, but only a 1:1 condition is shown. Showing the other concentrations would give a feel for dose dependent effects...

Author's response: as suggested, the overlay is now shown in Suppl. Figure S10b.

Reviewer 4, Comment 5: The authors' YEATS mutant studies and NCP study with unmodified histones versus free DNA support free DNA binding but seem to suggest that the bivalent nucleosomal DNA interaction and histone interaction is not as relevant as the cartoon in Figure 7 would suggest. The EMSA effects with the NCP with crotonylated lysine could also equally be explained by a monovalent interaction.

Author's response: we found that both interactions of AF9-YEATS, with H3K9cr and with DNA, are essential for the tight association with NCP. As shown in Figs. 5g and 6g, the YEATS mutants, defective in either H3K9cr binding or DNA binding, have a decreased capability to associate with H3K9cr-NCP.

The cartoon in Fig 7c has been revised to clarify that the basic patch binds to DNA.

Reviewer 4, Comment 6: Experimental. It was very difficult to evaluate conditions used in the experiments. If molar ratios are provided, I would suggest including the protein or DNA concentration in the figure legend rather than putting general conditions in the experimental section.

Author's response: as suggested, DNA and NCP concentrations for each experiment/figure are now specified in figure legends.

Reviewer #4 (Remarks to the Author):

I am now happy with the changes to the text and figures. I appreciate the authors attention to these points.